# Innovations in Diagnosis and Treatment of Coronary Artery Disease

**DOI:** 10.3390/diagnostics16010098

**Published:** 2025-12-27

**Authors:** Salaheldin Agamy, Sheref Zaghloul, Zahid Khan, Ahmed Shahin, Ramy Kishk, Ahmed Smman, Luciano Candilio

**Affiliations:** 1Cardiology Department, University Hospitals Birmingham, Hartlands Hospital, Birmingham B15 2GW, UK; salaheldin.agamy@nhs.net; 2Cardiology Department, Royal Berkshire Hospital, Reading RG1 5AN, UK; sheref.zaghloul@royalberkshire.nhs.uk; 3Bart’s Heart Centre, London and Queen Mary University, London E1 4NS, UK; drzahid1983@yahoo.com; 4Cardiology Department, Hampshire Hospitals NHS Foundation Trust, Basingstoke RG24 9NA, UK; ahmed.shahin@hhft.nhs.uk; 5Cardiology Department, Royal Free London NHS Foundation Trust, London NW3 2PS, UK; ramy.kishk@nhs.net (R.K.); ahmed.smman@nhs.net (A.S.)

**Keywords:** coronary heart disease, coronary revascularisation, coronary artery disease, ischemic heart disease, atherosclerosis, cardiovascular disease, non-invasive coronary diagnostic tests

## Abstract

**Background**: Coronary artery disease (CAD) remains a significant health challenge, placing a heavy burden on people and healthcare systems worldwide. **Objectives**: This narrative review aims to provide a comprehensive overview of recent advancements in the diagnosis, intervention, and pharmacological management of CAD, with a focus on emerging technologies shaping its future. **Methods**: This is a narrative review that synthesises information from diverse sources, including clinical trials, systematic reviews, meta-analyses, and preclinical studies, to provide a comprehensive overview of the current landscape and emerging trends in CAD management. The literature included in this review was sourced from original research articles and review papers published between January 2010 and December 2025. **Results**: Early detection has been transformed by non-invasive imaging, such as PCAT, and the addition of invasive and non-invasive FFR technology enables quicker and more accurate diagnoses. Biomarkers, such as high-sensitivity troponin, have further improved the precision of acute coronary syndrome detection, enhancing early intervention. In interventional cardiology, new-generation drug-eluting stents (DESs) have lowered restenosis rates, whereas robotic-assisted percutaneous coronary intervention (PCI) offers precision and reduced operator radiation exposure. Furthermore, the efficacy of drug-coated balloons (DCBs) has been established in the management of in-stent restenosis, and their application in de novo coronary lesions and bifurcation anatomy remains promising. Looking ahead, nanomedicine promises targeted plaque reduction and vascular repair, while 3D-bioprinted blood vessels offer durable, biocompatible grafts for surgical applications. Pharmacological developments, including modern cholesterol-lowering drugs, have also been crucial in achieving cholesterol targets. **Conclusions**: Despite significant advancements in diagnosis, intervention, and pharmacotherapy, several critical challenges remain, including the need for validated biomarkers and imaging modalities to identify vulnerable atheroma before symptoms arise. Continued research is essential to improve patient outcomes and address the global burden of CAD.

## 1. Introduction

Coronary artery disease (CAD) is the leading cause of death worldwide, responsible for over 19.2 million deaths in 2023 and one in three global fatalities [1]. Innovations in diagnosis and management have significantly improved outcomes, yet the burden persists, especially in low-resource settings.

The traditional understanding of CAD management centred on identifying and treating hemodynamically significant stenoses, often referred to as the “vulnerable lesion” paradigm. However, the last decade has witnessed a profound shift, moving the focus toward the “vulnerable patient”—an individual whose systemic risk factors and subclinical disease burden predispose them to an acute coronary event, regardless of the severity of a single lesion [2].

This paradigm shift is underpinned by landmark clinical trials that have redefined the role of revascularization in stable CAD. Studies such as the FAME 2 trial, the ORBITA trial, and the ISCHEMIA trial have collectively demonstrated that in patients with stable ischemic heart disease, an initial strategy of optimal medical therapy (OMT) often yields comparable long-term outcomes to an invasive strategy, particularly concerning the prevention of death and myocardial infarction [3,4,5,6,7]. Furthermore, the ORBITA-2 trial provided crucial evidence that while percutaneous coronary intervention (PCI) significantly improves angina symptoms and quality of life compared to a placebo procedure, the decision to intervene must be carefully balanced against the benefits of intensive pharmacological management [8].

This nuanced approach is further supported by trials in specific high-risk populations, such as the SENIOR-RITA trial, which showed that a routine invasive strategy did not significantly reduce cardiovascular death or myocardial infarction compared with a conservative strategy in older patients with non-ST-segment elevation myocardial infarction (NSTEMI) [9].

The recognition that up to 80% of acute coronary syndromes originate from plaques that were not severely stenotic (<50% luminal narrowing) has highlighted the critical importance of identifying and stabilising vulnerable plaques and controlling systemic risk [2]. Atherosclerosis is now understood as a chronic, systemic inflammatory disease driven by cumulative exposure to atherogenic lipoproteins and other risk factors. Therefore, the modern approach to CAD emphasises comprehensive risk factor control—including aggressive lipid lowering, blood pressure management, and the use of novel agents like SGLT2 inhibitors and GLP-1 receptor agonists—as the foundation of treatment [10,11].

This review brings together the most recent developments in CAD management, with an emphasis on three primary areas: advancements in diagnostics, progress in interventional cardiology, and breakthroughs in pharmacological treatments.

Despite these advancements, several critical challenges remain unaddressed in CAD management such as the need for validated biomarkers and imaging modalities that can identify vulnerable atheroma before symptoms arise see Figure 1.

## 2. Methods

This narrative review provides a comprehensive overview of recent advancements in the diagnosis, intervention, and pharmacological management of CAD, with a focus on emerging technologies shaping its future. A narrative review approach was employed to integrate evidence from diverse sources, including clinical trials, systematic reviews, meta-analyses, and preclinical studies, to provide a comprehensive overview of the current landscape and emerging trends in CAD management. This review is structured around four key domains—diagnostic innovations, advances in interventional cardiology, pharmacological breakthroughs, and future directions—that reflect the multifaceted evolution of CAD care.

The literature included in this review was sourced from original research articles and review papers published between January 2010 and December 2025. These references were identified through systematic literature search on PubMed, Scopus, Google Scholar, and institutional websites (e.g., The University of Edinburgh), using keywords such as “coronary artery disease,” “PCAT,” “artificial intelligence,” “fractional flow reserve,” “high-sensitivity troponin,” “lipoprotein(a),” “drug-eluting stents,” “robotic PCI,” “SGLT2 inhibitors,” “nanomedicine,” “3D bioprinting,” “ lipid-lowering agents, “ and “ drug-coated balloons.” Case reports, series, and editorials were excluded.

## 3. Diagnostic Innovations

### 3.1. Advanced Imaging Techniques

#### 3.1.1. High-Resolution CT Angiography for Early Plaque Detection

High-resolution coronary computerised tomography coronary angiography (CTCA), enabled by multidetector CT scanners, provides detailed imaging of the heart and coronary arteries, making it a class 1, evidence level A tool for detecting CAD [12]. While effective in identifying coronary calcium, plaque, and stenosis significance, its labour-intensive nature and demands highly skilled experts for image interpretation limit accessibility [13]. Advances in artificial intelligence (AI), particularly deep learning, enhance CTCA by accelerating analysis, detecting high-risk plaque features, and enabling precise risk stratification. AI also supports longitudinal studies on plaque progression and treatment efficacy, advancing personalised CAD management. This integration promises improved early detection, diagnosis, and patient outcomes [14].

#### 3.1.2. Pericoronary Adipose Tissue (PCAT)

Pericoronary adipose tissue (PCAT) is the fat surrounding coronary vessels within a radial distance equal to the vessel’s diameter, typically measured on CTCA. As a component of epicardial adipose tissue (EAT), PCAT is uniquely associated with atherosclerosis and cardiovascular risk factors [15].

Emerging evidence highlights its diagnostic potential in CAD through two key metrics: fat attenuation index (FAI) and PCAT volume. FAI, derived from CTCA, serves as a non-invasive biomarker for coronary inflammation, as vascular inflammation alters adipocyte composition, increasing water content and shifting CT attenuation. Elevated FAI reflects suppressed adipogenesis and reduced lipid content, while PCAT may also act as a local source of oxidised LDL, promoting plaque progression. Additionally, increased PCAT volume strongly correlates with coronary plaque presence, independent of BMI and other risk factors, making it a more specific marker than other fat depots [15].

Understanding variations in FAI and PCAT volume offers valuable insights for CAD diagnosis and risk stratification. Future research should validate PCAT as a prognostic marker and explore whether therapies aimed at modulating PCAT can improve outcomes in CAD patients [16].

#### 3.1.3. Non-Invasive Fractional Flow Reserve (FFR-CT) to Assess Blood Flow

FFR-CT is a computational post-processing technique that is applied to standard CT (CTCA) images. It employs artificial intelligence and computational fluid dynamics (CFD) to analyse hemodynamic parameters, aiding in the identification of ischemia-inducing coronary lesions. Unlike traditional CTCA, which provides only anatomical details, FFR-CT adds a functional perspective, enhancing diagnostic accuracy. By combining FFR-CT with plaque characterisation, clinicians can better stratify patient risk and make informed treatment decisions [17,18,19,20].

FFR-CT can effectively minimise unnecessary invasive procedures, thereby reducing the potential complications associated with them. Specifically, individuals with FFR-CT values exceeding 0.80 generally exhibit results similar to those without substantial coronary artery disease. Integrating FFR-CT into diagnostic workflows also contributes to lower healthcare expenses, mainly by reducing the need for invasive angiography. For example, data from the PLATFORM trial show that using CTCA combined with FFR-CT can substantially reduce costs in comparison to traditional approaches relying on immediate ICA [17,21].

Despite the ongoing advancements in FFR-CT technology, including improved AI models and real-time analysis, challenges such as image quality dependency, computational demands, and the need for widespread clinical validation remain. Further research is needed to expand its applicability to more complex cases, such as multi-vessel disease and in-stent restenosis assessment [22].

#### 3.1.4. Invasive Functional Assessment of Epicardial Stenosis Severity

Functional assessment of epicardial stenosis severity has become central to the debate on how best to guide coronary revascularization, particularly when angiographic estimates are inconclusive [20]. Evidence from landmark trials such as FAME 1 and 2, DEFINE-FLAIR, iFR-SWEDEHEART, R3F, and RIPCORD demonstrates that wire-based indices like fractional flow reserve (FFR) and instantaneous wave-free ratio (iFR) improve diagnostic accuracy compared with angiography alone, highlighting the poor correlation between visual stenosis severity and haemodynamic relevance. Importantly, intermediate lesions (40–90% non-left main, 40–70% left main) often show discordance, with a substantial proportion of moderate stenoses proving functionally significant and some severe stenoses not [7,23,24,25,26,27].

Debate persists regarding long-term outcomes: meta-analyses report a small excess in all-cause mortality with iFR compared to FFR, though both indices appear equally safe for deferral decisions. Systematic FFR in multivessel disease has not improved outcomes, reinforcing its role as a selective tool for intermediate lesions rather than routine application [23].

#### 3.1.5. Intravascular Imaging in the Detection of Vulnerable Plaque

Intravascular imaging modalities such as intravascular ultrasound (IVUS) and optical coherence tomography (OCT) have transformed the identification and characterisation of vulnerable plaques, a critical element in the pathogenesis of ACS. These plaques, particularly thin-cap fibroatheromas, are associated with a high risk of rupture, thrombosis, and subsequent myocardial infarction. Accurate detection of these lesions is therefore essential for patient risk stratification and for informing tailored interventional strategies [28,29].

IVUS operates on the principle of high-frequency ultrasound waves to visualise vessel wall architecture and plaque morphology. Its key advantage lies in its deep tissue penetration, typically around 10 mm, which enables comprehensive assessment of overall plaque burden and vessel remodelling. IVUS is highly effective in detecting positive remodelling and large necrotic cores within plaques. However, its moderate resolution, approximately 100 µm, limits detailed visualisation of thin fibrous caps. Furthermore, IVUS has limited capability in identifying microstructural features such as macrophage infiltration or microcalcifications, which are important markers of plaque vulnerability [29,30].

In contrast, OCT employs near-infrared light to produce cross-sectional images with significantly higher resolution, in the range of 10–20 µm. This superior resolution allows for the precise detection of thin-cap fibroatheromas and facilitates the identification of key microstructural features, including macrophage infiltration, microchannels, and microcalcifications. OCT is also valuable in evaluating stent apposition and neointimal coverage following percutaneous coronary intervention (PCI). Its primary limitation is a shallow penetration depth of 1–2 mm, which restricts visualisation of deeper plaque components and the external elastic lamina. Additionally, OCT imaging generally requires contrast injection, which may be contraindicated in patients with significant renal impairment [30,31].

When compared, IVUS and OCT offer distinct and complementary profiles. IVUS provides excellent assessment of vessel remodelling and global plaque burden, whereas OCT excels in the detection of fibrous cap thickness and microstructural details. For instance, identification of TCFA is highly reliable with OCT but poor with IVUS. Conversely, IVUS offers good assessment of lipid-rich cores, especially when combined with near-infrared spectroscopy (NIRS), while OCT’s shallow penetration limits its evaluation of large necrotic cores. Furthermore, macrophage infiltration is detectable with OCT but not reliably with IVUS [32].

Clinically, these differences guide their application. IVUS is optimal for planning PCI by providing a global assessment of lesion severity, vessel size, and remodelling. OCT, with its high-resolution capabilities, is particularly useful for identifying features associated with imminent plaque rupture, thereby enhancing risk prediction. In practice, the combined use of IVUS and OCT—sometimes integrated with adjunctive technologies like Near-Infrared Spectroscopy (NIRS)—can provide a more comprehensive plaque characterisation by merging the depth of penetration from IVUS with the high-resolution detail from OCT [28,32].

### 3.2. Biomarkers

#### 3.2.1. High-Sensitivity Troponin Assays for Early Detection of Myocardial Injury

High-sensitivity cardiac troponin (hs-cTn) assays have revolutionised the early detection of myocardial injury, particularly in diagnosing acute myocardial infarction (AMI). These assays enable the measurement of very low concentrations of cardiac troponins, allowing for the identification of minor myocardial injuries that were previously undetectable with conventional assays [33].

The advent of hs-cTn has further advanced both diagnostic and analytical performance, enabling the detection of troponin concentrations in a substantial proportion of asymptomatic, healthy individuals. This unique capability has opened new avenues for cardiovascular risk stratification in the general population. Accumulating evidence indicates that hs-cTn not only predicts future cardiovascular events but also responds to preventive pharmacological and lifestyle interventions, tracks in parallel with risk modification, and provides incremental prognostic value when integrated with established risk markers [34].

#### 3.2.2. Interleukin-6 (IL-6)

Interleukin-6 (IL-6) is a pro-inflammatory cytokine that plays a crucial role in the immune response and inflammation. IL-6 is involved in the activation of acute-phase proteins, such as C-reactive protein (CRP), and has been shown to promote endothelial dysfunction, which is a critical step in the development of atherosclerosis [35].

Elevated levels of IL-6 have been consistently associated with increased cardiovascular risk, including higher rates of myocardial infarction, stroke, and heart failure. For instance, a narrative review summarising data from prospective studies found that higher IL-6 levels correlate with adverse cardiovascular outcomes across diverse populations [35].

The relationship between IL-6 and CAD severity has also been explored through angiographic assessments. A study involving CAD patients stratified by IL-6 levels revealed that higher IL-6 concentrations are linked to greater disease severity, as indicated by higher Gensini scores and more extensive arterial involvement. This finding underscores IL-6′s role as a predictor of CAD progression [36].

While IL-6 extensively studied as a mediator of vascular inflammation, remains largely confined to scientific research and exploratory analyses, with limited translation into routine clinical practice. In contrast, hs TnI represents a clinically actionable biomarker with direct implications for preventive cardiology.

#### 3.2.3. Lipoprotein [Lp(a)]

Lipoprotein(a), or Lp(a), is a lipoprotein variant that consists of an LDL-like particle attached to a specific protein called apolipoprotein(a). Lp(a) is considered an independent risk factor for cardiovascular disease, particularly CAD. Unlike other lipoproteins, Lp(a) levels are primarily determined by genetics and remain relatively stable throughout an individual’s life [37,38].

Lp(a) has been shown to promote atherogenesis through several mechanisms, including the inhibition of fibrinolysis, promotion of endothelial dysfunction, and increasing the deposition of cholesterol in the arterial wall. Elevated Lp(a) levels have been linked to an increased risk of CAD, particularly in individuals who have a family history of premature cardiovascular disease [37,39,40].

Accumulating evidence positions IL-6 and Lp(a) as pivotal biomarkers in predicting CAD progression. Their measurement may refine risk stratification and enable personalised therapeutic strategies, particularly in patients with markedly elevated cholesterol, younger individuals at risk of premature disease, or those warranting more intensive intervention. Continued investigation is required to clarify their mechanistic roles and to inform the development of targeted therapies aimed at mitigating their pro-atherogenic effects.

#### 3.2.4. High-Sensitivity C-Reactive Protein

High-sensitivity C-reactive protein (hsCRP) has gained recognition as a residual risk factor in coronary artery disease, reflecting the systemic inflammatory burden that contributes to plaque destabilisation. Beyond its epidemiological association with recurrent cardiac events, hsCRP provides important biological insight into the mechanisms of plaque vulnerability. Elevated hsCRP levels are linked to endothelial dysfunction, macrophage infiltration, and matrix degradation, all of which promote the formation of thin-cap fibroatheromas and layered plaques. These processes highlight hsCRP not merely as a marker of risk, but as a surrogate for the inflammatory pathways that drive adverse remodelling within the coronary vasculature [41,42].

## 4. Advances in Interventional Cardiology for Coronary Artery Disease

The evolution of interventional cardiology has significantly improved the management of CAD, which is one of the leading causes of morbidity and mortality worldwide. This review focuses on three pivotal areas: drug-coated balloons, drug-eluting stents (DESs), and robot-assisted percutaneous coronary intervention (PCI). These technologies address complex clinical challenges and enhance outcomes for CAD patients by enabling precision, ensuring safety, and reducing complication rates [43].

### 4.1. Drug-Coated Balloons in CAD Management

Drug-coated balloons (DCBs) are a promising therapeutic modality for the management of CAD, providing targeted pharmacological intervention without the need for permanent vascular scaffold placement. Originally designed to address in-stent restenosis (ISR), their utility has expanded to include small-calibre vessels and bifurcation lesions, where conventional stenting may pose technical or long-term challenges [44,45].

#### 4.1.1. DCBs for ISR

ISR remains the most well-established indication for DCB therapy, primarily because it allows avoiding the use of multiple metallic stent layers. Among the available technologies, paclitaxel-coated balloons have undergone extensive evaluation in randomised controlled trials and now serve as the standard for emerging DCB platforms. Comparative studies have consistently demonstrated the superiority of DCBs over conventional balloon angioplasty for ISR management, with notable reductions in luminal narrowing and the need for repeat revascularisation procedures [45,46,47].

#### 4.1.2. DCB in Denovo Lesion

Early comparisons between DCBs and DESs for de novo small vessel lesions, such as in the PICOLETTO trial, revealed limitations of first-generation DCBs. It was primarily due to suboptimal drug delivery and inadequate vessel preparation [45]. However, subsequent randomised trials using improved paclitaxel-coated balloons demonstrated noninferiority to DES, supporting a DCB-only strategy in select cases [48,49,50,51].

#### 4.1.3. Future of DCBs

Bifurcation lesions pose procedural challenges and are associated with suboptimal long-term outcomes, making drug-coated balloons (DCBs) in the side branches an attractive alternative to conventional angioplasty. While observational data suggest improved patency and safety at 12 months, randomised trials remain limited and mixed in outcomes [51,52].

In addition to bifurcations, DCBs may offer distinct advantages in specific populations. Their scaffold-free design reduces vessel trauma and may lower the risk of thrombosis, potentially shortening the duration of dual antiplatelet therapy. This is particularly beneficial for patients at high risk of bleeding [53]. DCBs hold particular promise in patients with diabetes, where coronary disease tends to diffuse with longer lesions and DES underperforms with a higher incidence of ISR. DCBs may allow for shorter stented segments but longer areas of treatment, with the option of bail-out stenting if required, thereby decreasing the risk of restenosis [45].

### 4.2. Drug-Eluting Stents in CAD Management

#### 4.2.1. Historical Context

Bare-metal stents (BMSs) were the first breakthrough in CAD treatment, reducing acute vessel recoil and restenosis rates. However, the limitations of BMSs, including high rates of in-stent restenosis (up to 30%), led to the development of DESs. These stents combine a metallic scaffold, a polymer coating, and an antiproliferative drug to prevent neointimal hyperplasia [54].

#### 4.2.2. Modern Innovations

##### Thinner Strut Designs

Contemporary DESs are engineered with ultrathin struts, typically below 80 microns, enhancing deliverability through tortuous vessels, minimising vessel trauma, and accelerating endothelial healing. At the same time, clinical studies highlight their improved outcomes in patients with complex anatomies, including bifurcations and calcified lesions [55].

##### Biodegradable Polymers

Bioresorbable polymer coatings in drug-eluting stents, such as the Orsiro DES and Synergy stent, release their therapeutic drug over a predetermined period before degrading, leaving a bare-metal scaffold that reduces the long-term risk of late stent thrombosis [56,57].

##### Polymer-Free Stents

The BioFreedom stent exemplifies an innovative approach to stent design by employing microporous or nanoporous surfaces for drug delivery, eliminating the need for a polymer coating and thereby addressing concerns about polymer-induced inflammation and hypersensitivity [58].

##### Advanced Drugs

Antiproliferative Agents: Modern DESs employ sirolimus analogues such as everolimus, zotarolimus, and biolimus, which are more effective and better tolerated than earlier agents, such as paclitaxel [57].

#### 4.2.3. Clinical Benefits

DESs have significantly reduced restenosis rates to 2–10%, compared to 30% with BMSs. At the same time, biodegradable polymers lower the risk of late thrombosis, and faster endothelial coverage shortens the required duration of dual antiplatelet therapy, offering particular benefits for patients at high bleeding risk [57,59].

#### 4.2.4. Challenges

Neoatherosclerosis has been reported in approximately 30–40% of drug-eluting stents (DES) within 2–5 years after implantation, whereas in bare-metal stents (BMSs) it tends to occur later, often beyond 5 years [60]. Its development depends on multiple factors, including stent type (DES being more susceptible due to delayed endothelialisation), patient-related risk factors such as diabetes, hyperlipidaemia, smoking, and chronic kidney disease, and pharmacological influences such as discontinuation or inadequate antiplatelet therapy. Newer-generation DESs with biocompatible polymers appear to reduce but not eliminate this risk, highlighting the multifactorial nature of the phenomenon and the importance of long-term management strategies [61].

### 4.3. Robotic-Assisted Percutaneous Coronary Intervention

Robotic percutaneous coronary intervention (R-PCI) is an innovative method for PCI, enabling operators to remotely manipulate guidewires and catheter devices via advanced, precision-controlled technology [18,62].

#### 4.3.1. Key Features

##### Precision and Stability

Robotic systems, such as the CorPath GRX, provide sub-millimetre accuracy, essential for navigating complex lesions, including bifurcations and chronic total occlusions, enabling precise stent and balloon placement [62,63].

##### Radiation Protection

Operators work from a shielded console, minimising radiation exposure and alleviating the need for heavy lead aprons [18,63,64,65].

##### Remote Operation (Tele-Stenting)

Robotic PCI involves a collaborative process where vascular access is obtained by the in-lab cardiologist, and the robotic system is prepared with the necessary catheters and guidewires. Once the guide catheter is positioned in the coronary artery, the remote operator uses a robotic workstation to precisely advance the guidewire, balloon, and stent to treat the blockage. The procedure is supported by the in-lab team for imaging, contrast injections, and safety, ensuring accurate stent deployment with emergency backup available if needed [66].

#### 4.3.2. Clinical and Operator Benefits

Improved procedural accuracy, achieved through enhanced precision, minimises complications, such as malposition and edge dissection. This advancement has led to higher success rates, particularly in cases involving high-risk or anatomically challenging lesions [62]. Furthermore, operator ergonomics are significantly improved, reducing physical strain and occupational hazards, which contribute to a safer and more efficient procedural environment [66].

#### 4.3.3. Safety in Complex Lesions

Robotic PCI has proven to be highly effective in managing complex coronary lesions, as shown in the PRECISION and PRECISION GRX studies. These registries demonstrated that even challenging cases—such as calcified lesions, bifurcations, chronic total occlusions, and in-stent restenosis—can be treated safely and successfully with robotic platforms. The second-generation system, with enhanced guide catheter control and advanced software features, achieved higher technical success rates in these difficult scenarios, highlighting its ability to expand the scope of PCI while maintaining safety and precision in complex interventions [67].

##### Challenges

The adoption of robotic systems is hindered by their high cost, which makes them less accessible in low-resource settings. Additionally, their practical use requires extensive training and experience to achieve optimal outcomes. Current robotic systems also face limitations when addressing complex cases, such as multi-vessel disease and highly tortuous anatomies, further restricting their application in specific scenarios [68].

### 4.4. Shielding Systems for Radiation Protection

Interventional cardiology procedures expose medical personnel to significant ionising radiation, leading to occupational health risks such as cataracts, thyroid disorders, and musculoskeletal issues from heavy lead aprons [65,69]. To address these concerns, advanced fixed shielding systems have emerged as vital innovations. These systems create a protective barrier between the operator and the radiation source, aligning with the “As Low As Reasonably Achievable” (ALARA) principle and facilitating a shift towards a “lead-free” environment in cardiac catheterisation laboratories by minimising the reliance on traditional personal protective equipment [69].

Innovations in fixed shielding include comprehensive integrated systems, such as the Protego radiation shielding system (Image Diagnostics Inc., Fitchburg, MA, USA), and suspended-body shielding units, such as the Zero-Gravity system (BIOTRONIK, Berlin, Germany). The Protego system features an angulated upper shield, a lower shield, accessory side shields, flexible drapes with vascular access portals, and specialised arm boards, all designed to allow unimpeded C-arm motion while providing superior protection against radiation exposure [53]. The Zero-Gravity system, a 1 mm lead body shield suspended from the floor or ceiling, effectively reduces operator radiation exposure and alleviates the orthopaedic strain associated with lead aprons [69].

The adoption of these fixed shielding systems offers substantial benefits, including enhanced and consistent radiation protection and significantly reduced operator radiation exposure, well below recommended safety limits. They also mitigate the orthopaedic burden on interventional cardiologists, improving comfort, focus, and career longevity. These technologies represent a critical advancement in occupational radiation safety within interventional cardiology, ultimately benefiting both medical staff and patients [65,69].

### 4.5. Hybrid Coronary Revascularisation

Hybrid coronary revascularisation (HCR) is an emerging strategy that combines the durability of surgical grafting with the minimally invasive advantages of PCI. The standard technique involves an off pump left internal mammary artery (LIMA) graft to the left anterior descending artery (LAD) via minimally invasive direct coronary artery bypass (MIDCAB), supplemented by PCI to non-LAD vessels. This approach avoids full sternotomy and cardiopulmonary bypass while preserving the long-term benefits of arterial revascularisation. Optimal patient selection, guided by a multidisciplinary heart team, focuses on those with severe LAD disease and non-LAD lesions suitable for PCI. Evidence from observational studies and randomised trials supports the safety and feasibility of HCR in appropriately selected patients, although further large-scale randomised investigations are required to define its comparative role against standalone CABG or PCI [70].

### 4.6. Intravascular Lithotripsy (IVL)

Moderate-to-severe coronary artery calcification represents a significant challenge in PCI, occurring in approximately one-third of patients with stable coronary disease or acute coronary syndromes, and severe calcification in about 15% of cases. These calcified lesions are associated with lower procedural success, higher rates of periprocedural major adverse cardiovascular events (MACEs), and unfavourable long-term outcomes, including restenosis, stent thrombosis, and increased mortality. The rigidity and resistance of calcified plaques often make them difficult to cross and dilate, particularly in older patients and those with comorbidities such as diabetes and chronic kidney disease [71].

IVL has emerged as an innovative solution to this problem. Adapted from lithotripsy technology used for nephrolithiasis, IVL employs acoustic shock waves delivered through a balloon-based system to fracture calcium deposits within the vessel wall. This facilitates luminal gain and optimal stent expansion, improving procedural success and safety. The currently available IVL system (Shockwave Medical, Santa Clara, CA, USA) has demonstrated promising results in clinical studies, offering a controlled and effective approach to treating heavily calcified coronary lesions [72,73].

IVL has shown success in treating in-stent restenosis caused by calcified neoatherosclerosis and underexpanded stents in severely calcified lesions, where traditional devices like rotational atherectomy (RA) or scoring balloons are less effective due to the metallic scaffold. Compared to other modalities such as high-pressure ballooning, RA stentablation, or excimer laser, IVL effectively fractures calcium and facilitates stent expansion with relatively low risk [74].

## 5. Pharmacological Breakthroughs

### 5.1. Lipoprotein(a) Reduction

Elevated levels of lipoprotein(a) [Lp(a)] are recognised as an independent risk factor for coronary artery disease (CAD). Several therapeutic approaches have been investigated to reduce circulating Lp(a), with both conventional and novel agents showing promise [75].

Muvalaplin, an oral small molecule, has demonstrated reductions in Lp(a) levels when assessed through intact lipoprotein(a) and apolipoprotein(a)-based assays. Importantly, the drug has shown good tolerability in clinical studies. While these findings are encouraging, further trials are required to establish whether lowering Lp(a) with muvalaplin translates into improved cardiovascular outcomes. Evolocumab, a PCSK9 inhibitor, has also been shown to effectively lower Lp(a). Patients with higher baseline concentrations tend to experience greater reductions, and these individuals appear to derive more cardiovascular benefit from PCSK9 inhibition [75,76].

Beyond these established therapies, small interfering RNA (siRNA) agents are emerging as highly potent and long-acting strategies for Lp(a) reduction. Lepodisiran, developed by Eli Lilly, is designed to silence the LPA gene in hepatocytes, thereby reducing apolipoprotein(a) synthesis and circulating Lp(a). In the phase 2 ALPACA trial, lepodisiran achieved reductions of up to 94% after a single dose, with effects persisting for nearly a year. These findings highlight its potential as a durable therapy for patients with genetically determined elevations in Lp(a) [39].

Similarly, olpasiran, developed by Amgen, targets hepatic Lp(a) synthesis using siRNA technology. In the phase 2 OCEAN(a)-DOSE trial, olpasiran produced reductions exceeding 95% at doses of 75 mg or higher administered every 12 weeks. Remarkably, these effects were sustained for more than a year after the last dose. Such results underscore olpasiran’s potential as a long-acting therapeutic option for patients with elevated Lp(a) and established cardiovascular disease [40].

### 5.2. Anti-Inflammatory Therapies

#### 5.2.1. Monoclonal Antibodies

The CANTOS trial provided pivotal proof-of-concept that selective inhibition of interleukin-1β can attenuate recurrent cardiovascular events by targeting residual inflammatory risk, independent of lipid lowering. However, the absolute risk reduction was counterbalanced by a significant increase in fatal infections and sepsis, underscoring the immunosuppressive hazard of this therapeutic approach. These safety concerns, together with the high cost of canakinumab, have limited its clinical translation. Nevertheless, CANTOS catalysed a paradigm shift, reinforcing inflammation as a modifiable driver of atherosclerosis and stimulating subsequent investigations of alternative agents with more favourable safety profiles, such as low-dose colchicine and IL-6 pathway inhibitors [77,78].

#### 5.2.2. Colchicine

In LoDoCo2 trial, patients with chronic coronary artery disease, Colchicine 0.5 mg daily significantly reduced the risk of major cardiovascular events (chiefly MI and revascularization) by ~31%, with consistent reductions in the primary composite endpoint. This benefit was achieved on top of standard secondary prevention. However, it also revealed a potential uptick in non-cardiovascular mortality, with no increase in overall mortality [79].

#### 5.2.3. Methotrexate

Low-dose methotrexate (MTX) was rigorously evaluated in the Cardiovascular Inflammation Reduction Trial (CIRT), a large, randomised, double-blind, placebo-controlled study involving 4786 patients with established coronary artery disease and either type 2 diabetes or metabolic syndrome. Participants received weekly MTX (15–20 mg) plus folate versus placebo, with a median follow-up of 2.3 years. Despite high hopes based on MTX’s anti-inflammatory actions, CIRT showed no reduction in major adverse cardiovascular events (MACEs), defined as nonfatal myocardial infarction, nonfatal stroke, hospitalisation for unstable angina leading to revascularization, or cardiovascular death. Additionally, MTX failed to lower key inflammatory biomarkers—including interleukin 1β, interleukin 6, or C reactive protein—compared to placebo. The trial did demonstrate increased incidences of hepatic enzyme elevations, cytopenias, and non-basal-cell skin cancers among MTX recipients, reinforcing its unfavourable risk–benefit profile in this cardiovascular context [80].

### 5.3. Sodium–Glucose Cotransporter 2 (SGLT2) Inhibitors

Initially developed for the management of type 2 diabetes, SGLT2 inhibitors have shown cardiovascular benefits [81]. Sotagliflozin (Inpefa), a novel SGLT inhibitor, demonstrated a 23% reduction in heart attacks, strokes, and cardiovascular-related deaths compared to placebo. This positions sotagliflozin as a multifaceted therapeutic agent addressing interconnected health issues such as heart failure and diabetes [82].

Among the available SGLT2 inhibitors, dapagliflozin has emerged as a cornerstone therapy for cardiovascular care. The DAPA-HF trial established dapagliflozin’s efficacy in reducing the risk of worsening heart failure and cardiovascular death in patients with reduced ejection fraction, regardless of diabetes status [83]. More recently, the DAPA-MI trial evaluated dapagliflozin in patients with acute myocardial infarction without prior diabetes or chronic heart failure. The study demonstrated improved cardiometabolic outcomes, including reduced incidence of new-onset type 2 diabetes and favourable weight reduction. However, it did not show a statistically significant reduction in major adverse cardiovascular events compared to placebo [84].

Empagliflozin has demonstrated profound cardiovascular benefits. The EMPA-REG OUTCOME, EMPEROR-Reduced, and EMPEROR-Preserved trials collectively showed that Empagliflozin reduces MACEs and hospitalisation for heart failure across the full spectrum of patients, with and without diabetes [85,86].

### 5.4. New, Novel Lipid-Lowering Agents for Reducing Cardiovascular Risk

Advancements in lipid-lowering therapies beyond traditional statins for managing coronary artery disease (CAD), like proprotein convertase subtilisin/kexin type 9 (PCSK9) inhibitors (evolocumab and alirocumab), effectively reduce LDL cholesterol levels and lower cardiovascular risk, particularly in patients intolerant to statins [87,88]. Another breakthrough is inclisiran, an siRNA-based therapy that provides sustained LDL reduction with biannual dosing, enhancing patient adherence [88]. In a recent meta-analysis, inclisiran was shown to substantially reduce total cholesterol, ApoB, and non-HDL-C, respectively, by 37%, 41%, and 45% [89].

### 5.5. Glucagon-like Peptide-1 Receptor Agonists (GLP-1RAs)

GLP-1RAs, including liraglutide and semaglutide, have emerged as cornerstone therapies for cardiovascular risk reduction, with benefits extending beyond glycaemic control. Their anti-atherosclerotic actions are multifactorial: improving endothelial function via enhanced nitric oxide bioavailability, attenuating vascular inflammation through reduced adhesion molecule expression, stabilising plaques by decreasing lipid core size and strengthening fibrous caps, and exerting systemic benefits such as weight loss, blood pressure reduction, and improved lipid profiles [90].

Clinical outcome trials confirm these mechanistic benefits. The LEADER trial demonstrated liraglutide reduced major adverse cardiovascular events (MACEs) in high-risk patients with type 2 diabetes [91]. The SELECT trial showed semaglutide lowered MACEs by 20% in overweight/obese individuals with established CVD but without diabetes, highlighting glucose-independent heart protection [92]. The SOUL trial further validated oral semaglutide in reducing MACEs among patients with type 2 diabetes and CVD [93]. Collectively, these findings establish GLP-1RAs as integral to optimal therapy for atherosclerotic cardiovascular disease, irrespective of diabetic status.

## 6. Future Directions and Challenges

Atherosclerosis, a leading cause of heart attacks and strokes, is primarily driven by hypercholesterolemia and chronic inflammation within arterial walls [94]. Although statins and other standard therapies can slow disease progression, they often come with limitations such as side effects, high costs, and an inability to reverse established plaques [95]. In advanced cases, surgical interventions like coronary artery bypass grafting (CABG) are necessary, yet these procedures face significant hurdles, including limited graft availability and complications related to surgical trauma [96]. Given these challenges, innovative approaches such as nanomedicine and 3D bioprinting have emerged as promising alternatives for improving cardiovascular treatment.

### 6.1. Advancements in Nanomedicine

Nanomedicine, which utilises nanoparticles and nanocarriers to deliver drugs directly to atherosclerotic plaques, is revolutionising the management of cardiovascular disease [97]. These nanoparticles can be engineered to target inflamed arterial sites, increasing drug accumulation in plaques while minimising off-target effects [98].

Cholesterol clearance is a promising application of nanomedicine. Supramolecular nanotherapy has demonstrated the ability to dissolve cholesterol crystals, reducing plaque burden in preclinical studies [98]. Additionally, nanoparticles designed to inhibit macrophage-driven inflammation have been shown to stabilise plaques by blocking inflammatory signals [99]. Beyond plaque reduction, some nanosystems also facilitate vascular repair by restoring endothelial function, stabilising vessel walls, and preventing further narrowing of arteries [100].

Although these advancements are promising, translating nanomedicine from preclinical models to clinical applications remains challenging. Although lipid nanoparticle-based cholesterol efflux therapies have demonstrated efficacy in laboratory studies, only a few nanomedicine treatments have progressed to human trials [98]. Further research is needed to refine nanoparticle formulations, improve targeting precision, and ensure safety before widespread clinical adoption [101,102].

### 6.2. 3D-Printed Artificial Blood Vessels

In addition to nanomedicine, 3D bioprinting is emerging as a revolutionary technology in cardiovascular surgery. Researchers at the University of Edinburgh have developed a novel 3D bioprinting technique to fabricate artificial blood vessels that mimic natural veins [103]. The process involves printing a tubular gelatin hydrogel on a rotating spindle, followed by electrospinning an ultrathin biodegradable polyester nanofiber coating to enhance mechanical strength [104].

These 3D-printed vascular grafts offer several advantages over conventional grafts. By eliminating the need for vein harvesting, they reduce surgical trauma, post-operative pain, and the risk of infection [103,104]. Additionally, they are designed to be more biocompatible than synthetic small-diameter grafts, which often fail due to poor integration and thrombosis [104,105]. Future iterations of these grafts may incorporate patient-derived cells, creating living vessels that promote long-term integration and function [105].

Despite their potential, 3D-printed blood vessels remain in the preclinical stages. Ongoing studies are evaluating their durability, patency, and integration with native tissue in animal models. Before they can be used widely in cardiovascular surgery, large-scale human trials will be necessary to confirm their safety and efficacy [105].

### 6.3. Clinical Implications and Future Directions

Emerging technologies such as nanomedicine and 3D bioprinting have the potential to enhance current cardiovascular treatment strategies significantly. Targeted nanoparticle therapies have the potential to stabilise or shrink plaques, potentially reducing the need for invasive procedures [106]. Meanwhile, customised 3D-printed grafts may serve as durable and biocompatible replacements for diseased blood vessels, addressing a long-standing challenge in CABG surgery [103,104,105,106].

However, several hurdles remain before these technologies can be fully integrated into clinical practices. For nanomedicine, concerns about safety, scalability, and regulatory approval must be addressed [102]. Similarly, for 3D-printed blood vessels, extensive testing in human trials is required to establish long-term outcomes and reliability [106].

As research progresses, these advancements in nanomedicine and 3D bioprinting are expected to revolutionise cardiovascular therapy, offering new hope to patients with atherosclerosis and other vascular diseases.

## 7. Discussion

The diagnostic landscape has been transformed by non-invasive imaging, with machine learning-driven plaque analysis and Pericoronary Adipose Tissue (PCAT) attenuation enabling the identification of vulnerable, high-risk plaques. The focus in biomarkers has expanded from markers of acute injury (troponin) to those of systemic risk and inflammation, such as hs-CRP, hs-TnI, and emerging cytokines, which are directly linked to coronary pathobiology. In interventional cardiology, innovations such as IVL, advanced physiologic assessment (FFR/iFR), IVUS, OCT, and remote robotic PCI are enhancing procedural precision and safety, while the challenge of neoatherosclerosis is being actively investigated. Pharmacological breakthroughs, including siRNA therapies for Lp(a), broad-acting anti-inflammatory agents like colchicine, and the profound cardiorenal benefits of SGLT2 inhibitors and GLP-1 receptor agonists, are targeting residual cardiovascular risk with increasing efficacy. Looking forward, nanomedicine for targeted plaque therapy and 3D bioprinting of vascular grafts promise a future of regenerative and highly personalised medicine. Adding to this rapidly evolving landscape, early data from a Harvard-affiliated research group on a single-dose PCSK9-targeted gene-editing therapy (VERVE-102) demonstrate substantial, dose-dependent reductions in LDL-cholesterol of up to 69%. Although these findings remain preliminary and require validation in larger, long-term studies, they highlight the potential for durable, one-time interventions that could fundamentally reshape lipid management and long-term cardiovascular prevention [107].

The collective evidence from FAME, ORBITA, ISCHEMIA, and SENIOR-RITA underscores a central tension in the management of stable and elderly coronary artery disease: while physiology-guided or sham-controlled studies consistently demonstrate that percutaneous coronary intervention (PCI) provides meaningful relief of angina and improves quality of life, hard outcomes such as death and myocardial infarction remain largely unaffected when compared with optimal medical therapy (OMT). These findings reinforce the safety and efficacy of OMT as a foundational strategy, while positioning PCI as a tool primarily for symptom control rather than prognostic benefit in stable disease [3,5,6,7,8,9].

At the same time, the controversies surrounding trial design—ranging from early termination and “soft” endpoints in FAME 2, to short follow-up and intensive medical regimens in ORBITA, and the exclusion of high-risk populations in ISCHEMIA—highlight the complexity of translating trial results into practice. SENIOR-RITA further adds nuance by questioning the routine use of invasive strategies in older, frail patients. Taken together, these studies invite a more individualised approach: clinicians must balance the robust evidence for medical management with the undeniable symptomatic benefits of intervention, tailoring decisions to patient risk profiles, preferences, and quality-of-life considerations.

From the economic perspective, the burden of CAD is profound. In 2023, global healthcare spending on cardiovascular disease exceeded $1 trillion, with CAD accounting for a substantial proportion of direct and indirect costs [108]. Hospitalizations, long-term pharmacotherapy, and productivity losses due to disability contribute to this financial strain. Investment in early detection and population-based prevention strategies could hold substantial long-term savings and reduce reliance on high-cost interventions.

## 8. Conclusions

While technological and therapeutic advances have reshaped the landscape of CAD management, addressing the persistent clinical, economic, and accessibility challenges is vital to reducing the global burden of disease. Future efforts must prioritise early detection, equitable access, and cost-effective, personalised care to achieve sustainable improvements in cardiovascular outcomes.

## Figures and Tables

**Figure 1 diagnostics-16-00098-f001:**
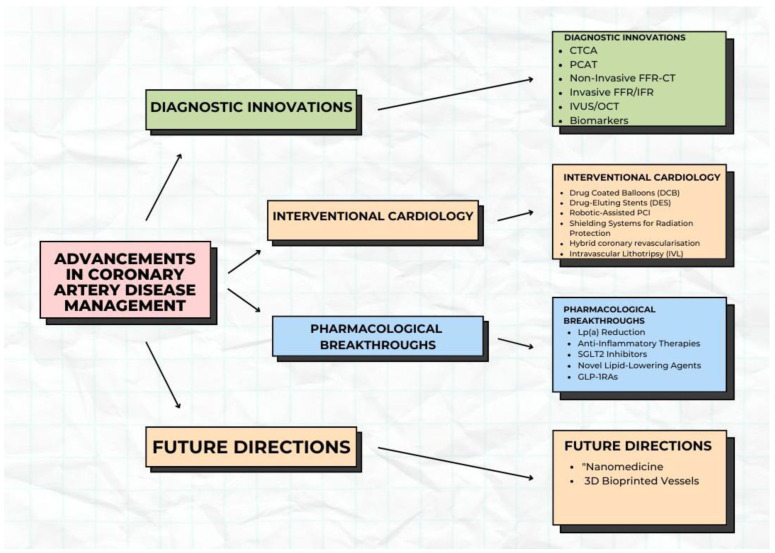
Subtopics under the advancement in coronary artery disease.

## Data Availability

No new data were created or analyzed in this study.

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
