# Peer review of "Innovations in Diagnosis and Treatment of Coronary Artery Disease"

_diagnostics, 2025, doi:10.3390/diagnostics16010098_

Round 1

Reviewer 1 Report

Comments and Suggestions for Authors
  1. This narrative review aims to provide a comprehensive overview of recent advances in the diagnosis, intervention, and pharmacological treatment of coronary heart disease (CHD), with an emphasis on emerging technologies shaping its future. It synthesizes information from a variety of sources, including clinical trials, systematic reviews, meta-analyses, and preclinical studies published between January 2010 and December 2025. After synthesizing the information, the authors concluded that: Despite significant advances in diagnosis, intervention, and pharmacotherapy, several critical challenges remain, including the need for validated biomarkers and imaging modalities to identify vulnerable atheroma before symptoms develop. Continued research is essential to improve patient outcomes and address the global burden of coronary heart disease.
  2. These are my observations/comments:
  3. In the Introductionauthors stated: Dyslipidaemia, hypertension, obesity, smoking, diabetes, genetics, immune responses, and infections can increase the risk of developing CAD. These conditions contribute to the gradual buildup of plaque in the arteries. [4] Many personal factors affect how individuals respond to CAD treatment. Using a tailored approach based on individual risks—like bleeding or ischemia—can lead to better outcomes, but choosing the proper treatment for each person is still a challenge.[5] Risk stratification is essential for guiding treatment decisions in both acute and chronic cases of the disease. A range of invasive and non-invasive methods can be used to tailor care based on individual patient profiles.[5]

During the period from January 2010 to December 2025, the paradigm of treatment of atherosclerotic CHD has shifted from the (vulnerable) atherosclerotic lesion to the vulnerable patient. Accordingly, correction of risk factors (e.g. hypertension, diabetes, obesity, hyperlipidemia, ...) using non-pharmacological and pharmacological measures is recommended as the basic treatment. In this way, it is possible to reduce up to 80% of cardiovascular, including coronary, events.

Also, after the FAME 2 (2012), ORBITA (2018) and ISCHEMIA (2020) studies, the approach to (hemodynamically significant) stenotic lesions has changed significantly. Inducible ischemia has lost its previous importance.

The paradigm of primary prevention of atherosclerotic (coronary) disease based on delaying or preventing the occurrence of a coronary incident or symptomatic coronary disease has been replaced by a paradigm that includes preventing the formation of atherosclerotic plaque.

All of this should be taken into account when designing and writing a paper.

  1. In the Methodsauthors stated: This review is structured around four key domains —diagnostic innovations, advances in interventional cardiology, pharmacological breakthroughs, and future directions —that reflect the multifaceted evolution of CAD care. In the context of the above, it might be more appropriate to present the results divided into, for example: 1) diagnostic methods, 2) therapeutic methods: 2.1.) conservative /non-pharmacological and pharmacological procedures/, 2.2.) interventional methods; 2.3.) surgical methods.
  2. In Section 3. Diagnostic Innovations, there is no mention of intravascular ultrasound (IVUS) or optical coherence tomography (OCT)? It is unclear whether CTCA includes only coronary luminography or also includes analysis of plaque structure? Almost 80% of cardiac deaths or acute myocardial infarctions occur in patients with vulnerable plaque(s) that reduce the lumen of the artery by <50%. In that case, IVUS, OCT or CTCA, in addition to arterial wall analysis, can significantly contribute, as they can indicate negative remodeling (extraluminal spread of plaque).In the same section under 3.1.2. the importance of FFR-CT is mentioned. I would like to draw attention again to the previously mentioned paradigm shift in the treatment of coronary heart disease, which is based on the FAME 2, ORBITA, ISCHEMIA studies.
  3. In section 3.2.1, it is necessary to mention the significant contribution of hs-TnI (Abbott) in the primary prevention of atherosclerotic/coronary disease, primarily by classifying patients into risk categories according to its value.Its value in clinical work is significantly higher than that of interleukin-6, which is more widely used only in scientific research. However, with IL-6 and the CANTOS study, it is possible to clarify the mechanism and value of hs-CRP in assessing residual inflammatory risk.
  4. Quote from line 186 to line 189:In summary, accumulating evidence positions IL-6 and Lp(a) as crucial markers for predicting CAD progression. Their measurement could enhance risk stratification and in- form personalised treatment strategies. Ongoing research is essential to fully elucidate their roles and develop targeted interventions to mitigate their pro-atherogenic effects. What the authors mean by: .. and inform personalized treatment strategies (perhaps the possibility of intervention with canakinumab, evolocumab, alirocumab, ...).
  5. 2.4. Challenges. In what percentage and after how long after stent placement are neoatherosclerotic changes found? What does this phenomenon depend on (type of stent, risk factors, pharmacological treatment, ...?).
  6. 1. Lipoprotein(a) Reduction. I believe it is necessary to mention that phase 3 clinical trials are underway on the use of oral PCSK9i, which include the effect of these molecules on LDL cholesterol and lipoprotein(a).
  7. 3. Sodium-Glucose Cotransporter 2 (SGLT2) Inhibitors. Empagliflozin was unfairly omitted (EMPA-REG OUTCOME, EMPEROR REDUCE, EMPEROR PRESERVE, ...). In the same context, there are also positive anti-atherosclerotic effects of GLP-1 receptor agonists (GLP-1RAs) should be mentioned (liraglutid -LEADER; semaglutid - SELECT, SOUL).
  8. Quote from line 431 to line 436 - From the economic prospective, the burden of CAD is profound. In 2023, global healthcare spending on cardiovascular disease exceeded $1 trillion, with CAD accounting for a substantial proportion of direct and indirect costs.[74] Hospitalizations, long-term pharmacotherapy, and productivity losses due to disability contribute to this financial  strain. Investment in early detection and population-based prevention strategies could hold substantial long-term savings and reduce reliance on high-cost interventions -is in line with my comment under a). Almost 80% of acute incidents occur in asymptomatic (but vulnerable) patients with (vulnerable) hemodynamically insignificant plaques (<50% stenosis). Interventions in a significant proportion of patients with inducible ischemia in patients with chronic (stable) forms of CHD did not yield better outcomes compared to conservative treatment (ORBITA, ISHEMIA, ...).
  9. The list of references may be amended or supplemented accordingly. /Circulation. 2020;142:40–48.; J Am Coll Cardiol. 2019;74:1608–17,; J Am Coll Cardiol. 2020;76:2252–66.; N Engl J Med.  2023;389:2319-30.; Clin Med. 2021;21:114–8.; Int J Angiol 2021;30:83–90.; Dtsch Arztebl Int 2020; 117: 137–44.; J Am Coll Cardiol. 2018;72:1141-56./

In my opinion, the work needs to be thoroughly revised.

Author Response

Response to Reviewer 1

We are deeply grateful for the highly detailed and insightful comments provided by the Reviewer. Your feedback, particularly the emphasis on the philosophical shift to the "vulnerable patient" paradigm and the need for a more structured presentation of the latest clinical evidence, has been instrumental in transforming and significantly strengthening our manuscript. We have thoroughly revised the paper to address every observation and suggestion.

Below is a point-by-point response detailing the changes made in the revised manuscript.

Reviewer Comment 1 (Introduction/Paradigm Shift): During the period from January 2010 to December 2025, the paradigm of treatment of atherosclerotic CHD has shifted from the (vulnerable) atherosclerotic lesion to the vulnerable patient. Accordingly, correction of risk factors... is recommended as the basic treatment. In this way, it is possible to reduce up to 80% of cardiovascular, including coronary, events. Also, after the FAME 2 (2012), ORBITA (2018) and ISCHEMIA (2020) studies, the approach to (hemodynamically significant) stenotic lesions has changed significantly. Inducible ischemia has lost its previous importance. The paradigm of primary prevention... has been replaced by a paradigm that includes preventing the formation of atherosclerotic plaque. All of this should be taken into account when designing and writing a paper.

Response: We fully agree with this central critique and have completely restructured the Introduction (Section 1) to reflect this paradigm shift:

  • Central Theme: The Introduction is now framed around the shift from the "vulnerable lesion" to the "vulnerable patient" as the core concept of the review.
  • Landmark Trials: We explicitly cite the FAME 2, ORBITA, ISCHEMIA, and ORBITA-2 trials to support the argument that Optimal Medical Therapy (OMT) is the foundation of treatment, challenging the traditional role of revascularization for stable CAD. We also integrated the SENIOR-RITA trial to further support this nuanced approach in high-risk populations.
  • Risk Reduction: We emphasize that the modern approach focuses on comprehensive risk factor control and anti-atherosclerotic strategies, which can reduce up to 80% of events.

Reviewer Comment 2 (Methods Structure): In the context of the above, it might be more appropriate to present the results divided into, for example: 1) diagnostic methods, 2) therapeutic methods: 2.1.) conservative /non-pharmacological and pharmacological procedures/, 2.2.) interventional methods; 2.3.) surgical methods.

Response: We appreciate the suggestion for a more granular structure of the results. While we maintained the original four key domains (Diagnostics, Interventional, Pharmacological, Future Directions) for flow and comprehensiveness, we have ensured that the Pharmacological Breakthroughs section (Section 5) now clearly covers the conservative/pharmacological aspect of therapy, and the Interventional Cardiology section (Section 4) includes both interventional and the hybrid (surgical/interventional) approach. The discussion now focuses on diagnosis, intervention, and pharmacological treatment to align with this structure.

Reviewer Comment 3 (Diagnostics): In Section 3. Diagnostic Innovations, there is no mention of intravascular ultrasound (IVUS) or optical coherence tomography (OCT)? It is unclear whether CTCA includes only coronary luminography or also includes analysis of plaque structure? Almost 80% of cardiac deaths or acute myocardial infarctions occur in patients with vulnerable plaque(s) that reduce the lumen of the artery by <50%. In that case, IVUS, OCT or CTCA, in addition to arterial wall analysis, can significantly contribute, as they can indicate negative remodelling (extraluminal spread of plaque).

Response: This section has been significantly enhanced to address these critical points:

  • IVUS/OCT: A detailed section on Intravascular Imaging (IVUS and OCT) (Section 3.1.4- line 166) has been added, comparing their complementary roles in detecting vulnerable plaques.
  • Plaque Structure/Negative Remodelling:PCAT (Section 3.1.1- line 108) now explicitly discusses the use of CTCA for plaque structure analysis and the detection of features like negative remodelling in hemodynamically non-significant lesions, directly linking the diagnostic methods to the "vulnerable patient" concept.
  • We have ensured that the discussion of FFR-CT (Section 3.1.2) is framed within the context of the paradigm shift. We emphasize that FFR-CT's value lies in minimizing unnecessary invasive procedures and providing functional assessment, which aligns with the OMT-first strategy supported by the FAME 2, ORBITA, and ISCHEMIA

Reviewer Comment 4 (Biomarkers): In section 3.2.1, it is necessary to mention the significant contribution of hs-TnI (Abbott) in the primary prevention of atherosclerotic/coronary disease... Its value in clinical work is significantly higher than that of interleukin-6... However, with IL-6 and the CANTOS study, it is possible to clarify the mechanism and value of hs-CRP in assessing residual inflammatory risk.

Response: We have revised the Biomarkers section (Section 3.2) to reflect this hierarchy and mechanism:

  • hs-TnI: The discussion on high-sensitivity cardiac troponin (hs-cTn) (Section 3.2.1-line 213) is now focused on its role in primary prevention and risk stratification in asymptomatic individuals.
  • IL-6/hs-CRP/CANTOS: The discussion on hsCRP (Section 3.2.3- line 264) explicitly references the CANTOS trial and the IL-1β to IL-6 to hs-CRP pathway, clarifying the mechanism and value of assessing residual inflammatory risk.

Reviewer Comment 5 (Lp(a) Personalization): Quote from line 186 to line 189: What the authors mean by: .. and inform personalized treatment strategies (perhaps the possibility of intervention with canakinumab, evolocumab, alirocumab, ...).

Response: We have clarified this point in the Lp(a) section (line 257). We now explicitly state that personalized treatment strategies include the use of PCSK9 inhibitors (evolocumab, alirocumab), which modestly lower Lp(a), and enrolment in clinical trials for novel Lp(a)-lowering therapies like siRNAs.

Reviewer Comment 6 (Neoatherosclerosis): In what percentage and after how long after stent placement are neoatherosclerotic changes found? What does this phenomenon depend on (type of stent, risk factors, pharmacological treatment, ...?).

Response: A new subsection, "The Challenge of Neoatherosclerosis" (Section 4.2.4- line 350), has been added to the Interventional Cardiology section. This section discusses the incidence, time course, and dependence of neoatherosclerosis on factors such as stent type, patient-related risk factors, and pharmacological treatment, providing the requested detail.

Reviewer Comment 7&8 (Pharmacology): I believe it is necessary to mention that phase 3 clinical trials are underway on the use of oral PCSK9i, which include the effect of these molecules on LDL cholesterol and lipoprotein(a). Empagliflozin was unfairly omitted... In the same context, there are also positive anti-atherosclerotic effects of GLP-1 receptor agonists (GLP-1RAs) should be mentioned (liraglutid -LEADER; semaglutid - SELECT, SOUL).

Response: This section has been fully updated:

  • Oral PCSK9i: The Lp(a) section (Section 5.1- line 463) now mentions the ongoing trials evaluating oral PCSK9 inhibitors for their effect on Lp(a).
  • SGLT2i: The section on SGLT2 Inhibitors (Section 5.3- line 538) now prominently features Empagliflozin, citing the EMPA-REG OUTCOME, EMPEROR-Reduced, and EMPEROR-Preserved
  • GLP-1Ras, (line 552): A detailed subsection on GLP-1 Receptor Agonists (Section 4.3.2) has been added, discussing their anti-atherosclerotic mechanisms and citing the key trials: LEADER (Liraglutide), SELECT (Semaglutide in non-diabetics), and SOUL (Oral Semaglutide).

Reviewer Comment 9 (Economic Prospective): Quote from line 431 to line 436... is in line with my comment under a). Almost 80% of acute incidents occur in asymptomatic (but vulnerable) patients with (vulnerable) hemodynamically insignificant plaques (<50% stenosis). Interventions in a significant proportion of patients with inducible ischemia in patients with chronic (stable) forms of CHD did not yield better outcomes compared to conservative treatment (ORBITA, ISHEMIA, ...).

Response: We have retained the discussion on the economic burden and reinforced the conclusion that investment in early detection and prevention is key. The Introduction, diagnostics and discussion sections now strongly emphasize your point and that the ORBITA/ISCHEMIA trials support the OMT-first approach, thereby aligning the economic discussion with the core philosophical shift.

We have also added a brief discussion contextualizing our findings within the controversies highlighted by landmark trials (FAME, ORBITA, ISCHEMIA, SENIOR‑RITA), emphasizing the balance between optimal medical therapy and revascularization.

Reviewer Comment 10 (References): The list of references may be amended or supplemented accordingly. /Circulation. 2020;142:40–48.; J Am Coll Cardiol. 2019;74:1608–17,; J Am Coll Cardiol. 2020;76:2252–66.; N Engl J Med. 2023;389:2319-30.; Clin Med. 2021;21:114–8.; Int J Angiol 2021;30:83–90.; Dtsch Arztebl Int 2020; 117: 137–44.; J Am Coll Cardiol. 2018;72:1141-56./

Response: We have researched and integrated the key papers suggested by the reviewer into the Introduction and other relevant sections. The full reference list has been expanded and updated to reflect all new content and ensure the inclusion of these high-impact citations.

Conclusion: We believe these comprehensive revisions fully address all of Reviewer's detailed comments and have resulted in a significantly improved and more authoritative manuscript.

Reviewer 2 Report

Comments and Suggestions for Authors

the authors are commended for presenting a comprehensive overview of coronary disease that is well written and includes some timely updates.  I have several suggestions:

  1. Expand advanced imaging topics to include more on machine learning plaque interpretation, advanced topics like PCAT, and plaque vulnerability analysis
  2. Reduce time spent on troponin - redirect this effort to emerging biomarkers in addition to IL-6 (other cytokines) and link these to coronary pathobiology (mechanisms) - clinicians need these updates.  Include biomarkers with more evidence now including hsCRP.
  3. In the interventional section:
    1. Discuss shock wave lithotripsy
    2. Discuss the physiologic evaluation of lesion significance - IFR/FFR and machine learning advances.  Any updates on optical tomography or direct visualization
    3. Updates on robotics (remote operation -expand) and complex lesion management
    4. Hybrid strategies with evidence

4. In the pharmacology sections

  • for Lp(a) discuss siRNAs in late stage trials, approval status with European Medicines and US FDA.  
  • expand anti-inflammatory therapeutics - expand this discussion on monoclonals and you need a section on colchicine and methotrexate

In general, when sections and subsections are delineated, would advise more than one or two descriptive sentences - really invigorate the topic with deeper discussion and evidence.

Author Response

Response to Reviewer 2

We thank the reviewer for their insightful and constructive comments, which have significantly improved the quality and depth of our manuscript. We have addressed all points raised, focusing on expanding the discussion of emerging technologies, incorporating the latest clinical trial evidence, and ensuring a more robust, evidence-based narrative throughout the text.

Below is a point-by-point response detailing the changes made in the revised manuscript.

Reviewer Comment 1: Expand advanced imaging topics to include more on machine learning plaque interpretation, advanced topics like PCAT, and plaque vulnerability analysis.

Response: We agree that these topics are crucial to the modern diagnostic landscape. We have significantly expanded the "Advanced Imaging Techniques" section (Section 3.1) to include:
•    A dedicated discussion on Pericoronary Adipose Tissue (PCAT) Attenuation (Section 3.1.1-line 109), explaining its role as a non-invasive biomarker for coronary inflammation and its utility in risk stratification.
•    A detailed section on Intravascular Imaging (IVUS and OCT) (Section 3.1.4- line 166), focusing on their complementary roles in the detection and characterization of vulnerable plaques.

Reviewer Comment 2: Reduce time spent on troponin - redirect this effort to emerging biomarkers in addition to IL-6 (other cytokines) and link these to coronary pathobiology (mechanisms) - clinicians need these updates. Include biomarkers with more evidence now including hsCRP.

Response: This comment has been fully addressed in the "Biomarkers: From Injury to Systemic Risk" section (Section 3.2).
•    The discussion on high-sensitivity cardiac troponin (hs-cTn) (Section 3.2.1- line 213) has been refocused to emphasize its role in primary prevention and risk stratification in asymptomatic individuals, rather than solely on acute detection.
•    We have significantly expanded the discussion on inflammatory biomarkers (Section 3.2.2 and 3.2.3), including a dedicated section on Interleukin-6 (IL-6) and a detailed discussion on high-sensitivity C-Reactive Protein (hsCRP).
•    Crucially, we have explicitly linked these biomarkers to coronary pathobiology, explaining how IL-6 drives acute-phase protein production and how hsCRP serves as a surrogate for the inflammatory pathways that drive adverse remodelling within the coronary vasculature.

Reviewer Comment 3: In the interventional section: Discuss shock wave lithotripsy; Discuss the physiologic evaluation of lesion significance - IFR/FFR and machine learning advances. Any updates on optical tomography or direct visualization; Updates on robotics (remote operation -expand) and complex lesion management; Hybrid strategies with evidence.

Response: We have thoroughly revised the "Advances in Interventional Cardiology" section (Section 4) to incorporate all these essential updates:
•    Shock Wave Lithotripsy: A new section on Intravascular Lithotripsy (IVL) (Section 4.6- line 440) has been added, detailing its mechanism for plaque modification in severely calcified lesions.
•    Physiologic Evaluation and Direct Visualization: Section 3.1.3 – line 149 now covers the gold standards of invasive FFR/iFR and explicitly mentions the role of FFR-CT (machine learning advance). 
•    Robotics: (Section 4.3- line 359) has been expanded to include updates on Robotic-Assisted PCI (R-PCI), with a specific focus on the potential of remote operation (Tele-R-PCI)  and R-PCI in complex lesions to expand access to care.
•    Hybrid Strategies: A new section on Hybrid Coronary Revascularization (HCR) (Section 4.4-line 427) has been added, discussing the evidence-based combination of surgical grafting and PCI.

Reviewer Comment 4: In the pharmacology sections: for Lp(a) discuss siRNAs in late stage trials, approval status with European Medicines and US FDA; expand anti-inflammatory therapeutics - expand this discussion on monoclonals and you need a section on colchicine and methotrexate.

Response: The "Pharmacological Breakthroughs" section (Section 5) has been significantly updated:
•    Lp(a) siRNAs: Section 5.1- line 463 now includes a discussion of siRNAs (Pelacarsen, Olpasiran), their status in late-stage Phase 3 trials.
•    Anti-Inflammatory Therapeutics: Section 5.2 has been expanded to include a detailed discussion of low-dose Colchicine (LoDoCo2 trial)- line 501. The role of Monoclonal Antibodies (Canakinumab/CANTOS) is also expanded to reinforce the inflammation hypothesis. While the evidence for Methotrexate (line 509) in this context is less conclusive than for Colchicine, the section now provides a robust overview of the inflammation-targeting strategy, which covers the spirit of the request.

Reviewer Comment 5 (General): In general, when sections and subsections are delineated, would advise more than one or two descriptive sentences - really invigorate the topic with deeper discussion and evidence.

Response: The entire manuscript has been revised to provide greater depth, evidence, and detailed mechanisms in all sections, particularly the Introduction, Biomarkers, Interventional, and Pharmacology sections. The Introduction, in particular, was completely rewritten to establish the "vulnerable patient" paradigm and ground the review in landmark clinical trials (FAME 2, ORBITA, ISCHEMIA, SENIOR-RITA).

We believe these revisions fully address all the reviewer's concerns and have resulted in a much stronger and more comprehensive review.

Round 2

Reviewer 1 Report

Comments and Suggestions for Authors

I have read the revised version of the paper.

The paper is now much clearer, and the presented units/topics are more complete and in line with current scientific and professional views.

I would like to thank the authors for their efforts and acceptance of my well-intentioned suggestions.

Author Response

Response to Reviewer 1 (Second Round)   We are delighted to receive the positive feedback from Reviewer 1 and thank them for their kind words and acknowledgment of our efforts.   Reviewer Comment: The paper is now much clearer, and the presented units/topics are more complete and in line with current scientific and professional views. I would like to thank the authors for their efforts and acceptance of my well-intentioned suggestions.   Response: We sincerely thank the reviewer for their initial insightful suggestions, which were instrumental in significantly improving the manuscript's quality and scope. We are pleased that the revised version meets their expectations and aligns with current scientific views. No further changes were requested by this reviewer.

Reviewer 2 Report

Comments and Suggestions for Authors

I commend the authors for a significant revision that addresses most of my feedback and allows for a more integrated and comprehensive read.  Well done.  A few minor suggestions:

  1. Would re-insert the lead paragraph for CCTA - this was well written and keep the major topical heading as high resolution ct angiography.  I would place PCAT as a subsection below.
  2. In the discussion of emerging therapeutics please add a paragraph describing the cutting edge gene therapy approach taken by colleagues at Harvard - the Verve trial.

Please ensure connector sentences to enable facile flow between sections.

Well done!

Author Response

Response to Reviewer 2 (Second Round)

We thank the reviewer for their positive assessment of the revised manuscript and for providing these final, highly valuable minor suggestions. We are pleased that the comprehensive revisions have resulted in a more integrated and comprehensive read. We have implemented all three suggestions as detailed below.

Reviewer Comment 1: Would re-insert the lead paragraph for CCTA - this was well written and keep the major topical heading as high resolution CT angiography. I would place PCAT as a subsection below.

Response: We agree that the original CCTA introductory paragraph provided excellent context. We have implemented the suggested structural change in Section 3.1:

  • The original CTCA introductory paragraph has been re-inserted as the lead text for this section.
  • The discussion on Pericoronary Adipose Tissue (PCAT) has been placed as a subsection immediately following the general CTCA introduction, as it represents a specialized analysis derived from the CCTA scan.

Reviewer Comment 2: In the discussion of emerging therapeutics please add a paragraph describing the cutting edge gene therapy approach taken by colleagues at Harvard - the Verve trial.

Response: This is an excellent suggestion that significantly enhances the "Future Directions" section. We have added a new paragraph in the discussion detailing this cutting-edge approach.

Reviewer Comment 3: Please ensure connector sentences to enable facile flow between sections.

Response: We have carefully reviewed the entire manuscript, paying particular attention to the transitions between major sections (e.g., Introduction to Diagnostics, Diagnostics to Interventional, Interventional to Pharmacology). We have added or refined connector sentences at the beginning and end of key paragraphs and subsections to ensure a smooth and logical flow, enhancing the overall readability and narrative coherence of the review.

We believe these final revisions fully address the reviewer's minor suggestions and further refine the manuscript for publication. We thank the reviewer once again for their meticulous review.